RESEARCH ARTICLE *Mem Inst Oswaldo Cruz*, Rio de Janeiro, Vol. *121*: e250114, 2026   1|11

# Comprehensive identification and co-expression analysis of long non-coding RNAs across eight timepoints of *Schistosoma mansoni* life cycle

**Flávia Arêdes-Rocha¹, Cristian Chaparro², Christoph Grunau², Renata Guerra-Sá³/⁺**

¹Universidade Federal de Ouro Preto, Núcleo de Pesquisas em Ciências Biológicas, Programa de Pós-Graduação em Biotecnologia, Ouro Preto, MG, Brasil
²Univ Montpellier, Univ Perpignan Via Domitia, Centre National de la Recherche Scientifique, Institut Français de Recherche pour l'Exploitation de la Mer, Interactions Hôtes-Pathogènes-Environnements, Perpignan, France
³Universidade Federal de Ouro Preto, Instituto de Ciências Exatas e Biológicas, Departamento de Ciências Biológicas, Laboratório de Bioquímica e Biologia Molecular, Ouro Preto, MG, Brasil

**BACKGROUND** Long non-coding RNAs (lncRNAs) are important regulatory molecules that can be considered potential biomarkers for schistosomiasis. However, the identification and characterisation of these molecules in such complex organism as *Schistosoma mansoni*, remains very challenging.

**OBJECTIVES** This study aimed to identify novel lncRNAs in *S. mansoni* using RNA-seq libraries representative of six developmental stages (eggs, miracidia, sporocysts, cercariae, schistosomula, and juveniles).

**METHODS** A pipeline for lncRNAs identification was defined and applied to 41 RNA-seq libraries of eggs, miracidia, 1-day, 5-days, 32-days sporocysts, cercariae, schistosomula, and juveniles of *S. mansoni*. Novel identified lncRNAs and annotated mRNAs were submitted to a weighted co-expression network analysis (WGCNA) to establish lncRNA-mRNAs correlation.

**FINDINGS** We identified 1,082 novel lncRNAs, mostly presenting stage-specific expression. Co-expression analysis demonstrate that MSTRG.5305.1 can potentially target MEGs and tetraspanin, essential for juvenile worm survival, making it a promising candidate for further functional studies.

**MAIN CONCLUSIONS** These findings expand the current catalog of *S. mansoni* lncRNAs and provide new insights into their potential roles in parasite development and host adaptation. Further functional validation could reveal new molecular targets for schistosomiasis control.

Key words: schistosomiasis - *Schistosoma mansoni* - lncRNAs - co-expression analysis - parasite development

Schistosomes are helminths of the *Schistosoma* genus, responsible for schistosomiasis, a neglected tropical disease. Human infection occurs through contact with freshwater contaminated by *Schistosoma* species, primarily *Schistosoma mansoni*, *Schistosoma haematobium*, and *Schistosoma japonicum*. Together, these three species are responsible for more than 250 million infections worldwide, making schistosomiasis the second most impactful parasitic disease in terms of socioeconomic burden, surpassed only by malaria.[1,2]

Like other helminths, schistosomes exhibit a complex digenetic cycle. Asexual reproduction occurs in the intermediate host, freshwater snails of the Planorbidae family, through sporocyst development, while sexual reproduction of adult worms takes place in the definitive host. Infection occurs through free-living larval stages: miracidia hatch from eggs and infect snails, where they differentiate in sporocysts; cercariae are released from snails after sporocyst maturation, penetrate the skin of vertebrate hosts and differentiate into schistosomula before becoming adult worms.[1]

Schistosomes are the only dioecious organisms among the trematodes, with heterogametic females (ZW) and homogametic males (ZZ). In addition to the pair of sex chromosomes, the worm possesses seven pairs of autosomes.[3]

Efforts to sequence the *S. mansoni* genome, the most widespread and studied species, began in 1994. The first version of the genome was published in 2009, but it was still incomplete due to the presence of numerous scaffolds aligning to multiple genomic regions, particularly in repetitive sequences (~40% of the genome). Multiple updated versions have since been released, with the most recent version comprising 391 Mb of sequenced data and 10,960 annotated gene transcripts.[4,5]

On the other hand, the transcriptome of *S. mansoni* was released in 2003.[6] Following the Encyclopedia of DNA Elements (ENCODE) project, non-coding RNAs

Financial support: CNPq, CAPES (Finance Code 001), FAPEMIG (APQ-03158-24).
+ Corresponding author: rguerra@ufop.edu.br | https://orcid.org/0000-0003-2486-0534

**Handling editor:** Adeilton Alves Brandão | https://orcid.org/0000-0001-5877-607X

(ncRNAs) gained attention as key regulatory molecules. LncRNAs are typically defined as RNAs longer than 200 nucleotides with little or no protein-coding capability. They regulate a wide range of cellular processes, including transcriptional and post-transcriptional modifications, through interactions with DNA, RNA, and proteins. These molecules can be classified according to the nearest protein-coding gene (PCG) as follows: intronic, when nested within introns; intergenic, when located between PCGs without overlapping; and antisense, when overlapping fully or partially with the sense PCG.[7,8]

The novelty and heterogeneity of lncRNAs make their identification particularly challenging, especially in non-model organisms. As a result, the development of computational tools for lncRNA identification is ongoing, with frequent updates and new releases of relevant software.[9,10]

In *S. mansoni*, more than 16,000 lncRNAs have been described to date.[11,12,13,14,15] Despite the growing number of studies exploring lncRNAs in schistosomiasis, the high complexity of their life cycle and genome can lead to misclassified molecules. Given the pivotal role of gene expression regulation across the parasite's distinct hosts and environmental conditions throughout its life cycle, lncRNAs may be key players in the phenotypic plasticity of *S. mansoni*.[9] Understanding which molecules are involved in these regulatory processes, and how they function, could help explain molecular mechanisms that remain poorly understood, highlighting the need for continued investigation into the functions and biological significance of lncRNAs.

Following the recent release of *S. mansoni* genome version 10, we focused here on identifying novel lncRNAs in *S. mansoni*. To improve the coverage of lncRNAs identification, RNA-seq libraries used in this project were highly representative of *S. mansoni* cycle: eggs, miracidia, sporocysts, cercariae, schistosomula, and juveniles. Newly identified lncRNAs were submitted to a co-expression analysis with annotated mRNAs to highlight promising transcripts.

## MATERIALS AND METHODS

*Data availability* - To identify lncRNAs in *S. mansoni*, RNA-seq samples from different stages of the parasite's life cycle covering eight distinct time points, were obtained from the National Centre for Biotechnology Information (NCBI): eggs, miracidia, sporocysts, cercariae, schistosomula, and juveniles (NCBI BioProject PRJEB32839).[16] All RNA-seq datasets originate from RNA extracted from mixed samples, except for juveniles, which were collected from a mixed infection and separated into three female and three male groups for library construction. However, the female and male datasets were merged, as sex-specific molecules were not the focus of this study. The eggs, 32-day sporocysts, and juvenile data originate from samples originally collected and purified from host tissues, whereas the other stages were cultivated and transformed *in vitro*. Details on sample collection and library construction can be found in PRJEB32839, and the corresponding codes are available in Supplementary data (Table I).

The *S. mansoni* genome FASTA and annotation files were downloaded from WormBase Parasite (release 19, 2024) in the most recent version (v 10.0).

*Bioinformatic analysis* - RNA-seq data were analysed using the Galaxy instance of the Interactions-Hôte-Patogène-Environment laboratory (IHPE). Low-quality reads were removed with Trimmomatic (v0.39)[17] using: Illumina adapter removal step, HEADCROP 12, SLIDINGWINDOW 5:20, LEADING and TAILING 3, and minimum reads size of 80 bp. Quality inspections were made with FastQC (v0.74).

Trimmomatic *fastq* outputs were mapped against *S. mansoni* reference genome v10 using RNA STAR (v2.7.11a)[18] setting — outSAMstrandField intronMotif. The resulting BAM files were assembled with StringTie (v2.2.3)[19] in a *de novo* approach using -rf reverse library parameter, and merged into a single GTF file for each developmental stage.

Non-redundant reconstructed transcripts were used as input for the Flexible Extraction of Long Non-Coding RNAs (FEELnc; v0.2.1)[20] software, a reliable tool for non-model organisms. In the first step, the filtering module of FEELnc removed transcripts overlapping protein-coding exons in sense, monoexonic and biexonic transcripts with small exons, and transcripts shorter than 200 nucleotides. Coding potential cut-off was generated by *shuffle* mode in FEELnc codpot module, which uses a random forest algorithm to establish the value. The putative lncRNAs distinguished by FEELnc codpot were merged in a single GTF file and compared with previously identified *S. mansoni* lncRNAs. Finally, the lncRNA candidates were analysed using BLASTx against protein databases, including NCBI (nr), Pfam, and SwissProt/UniProt, to remove any molecules with more than 30% sequence similarity to known proteins.

*Weighted gene co-expression network analysis (WGCNA) and Gene Ontology (GO) analysis* - A scale-free co-expression network analysis was performed to identify correlation between lncRNAs and mRNAs. First, raw reads for each annotated mRNAs and newly identified lncRNA transcripts were counted using Salmon (v1.10.1).[21] Then, the raw reads counts were submitted into a "goodSamplesGenes" analysis of the WGCNA (v1.73)[22] R package to identify outliers. After removal of samples and transcripts outliers, the counts were submitted to a variance stabilisation with DEseq2 (v1.44.0) vst method.[23] Based on scale-free topology criteria, a soft-thresholding power ($\beta$) of 22 was selected. The transcripts were clustered into modules, represented by colours, using the *blockwiseModules* from WGCNA, and visualised in a hierarchical clustering dendrogram.

A correlation test of each module and the *S. mansoni* developmental stages was performed using *corPvalueStudent* from WGCNA. The adjacency matrix was calculated and converted into a signed Topological Overlap Matrix (TOM), to determine the association level between transcripts.

Last, to identify lncRNAs as hubs in the co-expression network, the transcripts were ranked by Module

Membership (MM) determination with signed *kME* function. A gene significance (GS) was established according to correlation between the transcripts and the traits. LncRNAs from the yellow module with MM > 0.75 and GS > 0.5 were considered hubs and visualised with their co-expressed mRNAs in a subnetwork in Cytoscape (v3.10.1). A GO enrichment was performed for co-expressed mRNAs, using gplofiler2 R package for *S. mansoni* Gene Ontology Biological Process (GO:BP), Molecular Function (GO:MF), Cellular Component (GO:CC), KEGG pathways, and Reactome pathways (REAC) bases. Only significantly enriched terms (adjusted p < 0.05) were retained for visualisation.

*LncRNA expression in adult worm and cercariae* - RNA was extracted and purified from both adult worms and cercariae (~50 specimens) samples using the SV Total RNA Isolation System (Promega™), which includes on-column DNase I treatment. RNA quantity and purity were assessed using a spectrophotometer, and RNA integrity was verified by agarose gel electrophoresis. cDNA was synthesised using 1.5 µg of total RNA in the High-Capacity cDNA Reverse Transcription Kit (Applied Biosystems).

Primers for hub lncRNAs of the yellow module were designed using Gene Runner software (desktop version 6.5.52), considering the parameters of 18-20 nucleotides (nts) of length, 50-65% GC content, and 70-110 nts of amplicon length, which did not present any stable dimers or loops, generating: MSTRG.5305.1 forward 5' ACCATGGGTGACTATTGC 3' and reverse 5' TGCCTGCACAACTGAGTTC 3'; MSTRG.6595.5 forward 5' GGAAGTCGCTTTGTGTTTG 3' and reverse 5' TATGTGCACTCGTTTCTTG 3'; MSTRG.804.4 forward 5' CAAGTTTGTGGCCTAAAGG 3' and reverse 5' TTTCGCGTACGTGTAACG 3'.

The designed primers were used for the amplification reactions containing 3 µL of 10X diluted cDNA, 2 µL of oligonucleotides [300 nM] and the GoTaq® qPCR Master Mix (Promega) kit, following the manufacturer's recommendations. Gene expression was assessed in three biological replicates, and transcript levels were normalised against the endogenous control Eukaryotic Translation Initiation Factor 4E (EIF4E), using the primers forward 5' TGTTCCAACCACGGTCTCG 3' and reverse 5' TCGCCTTCCAATGCTTAGG 3', applying the $2^{-\Delta Cq}$ method.

*Statistical analysis* - LncRNA expression data were analysed using the software R (version: 4.3.2) and its graphical interface R Studio (version: 2024.09.1). Statistical analyses were performed using functions from the base R package stats, where a Shapiro-Wilk test was applied to verify data distribution, and group comparisons were carried out using the Mann-Whitney test (p < 0.05).

## RESULTS

*Schistosoma mansoni LncRNAs identification* - To enhance the scope and reliability of lncRNA identification, we analysed RNA-seq data from eight different time points (eggs, miracidia, 1-, 5-, and 32-days sporocysts, cercariae, schistosomula, and juveniles) in the *S. mansoni* life cycle, comprising a total of approximately 1.3 billion reads. Alignment resulted in 907 million mapped reads, with alignment rates between 73.66% and 87.63%. The only exception was the 32-day sporocyst stage, which had a mean alignment rate of only 32.27%, likely due to host RNA contamination.

From a total of 110,674 reconstructed transcripts, 16,791 were classified as lncRNAs, of which 1,082 [Supplementary data (Table II)] had not been previously described as lncRNAs in *S. mansoni*.[11,12,13,14] To ensure that host contamination did not affect subsequent analyses — particularly given the likelihood of contamination in the 32-day sporocyst samples — we aligned the identified lncRNAs against the *Biomphalaria glabrata* genome, the intermediate host of *S. mansoni*. No alignments were detected, indicating the absence of host-derived sequences or homologous regions. The majority of identified lncRNAs were located on chromosomes 1 and Z (Fig. 1A). Most of these molecules were approximately 1,500 bp in length, with the majority being shorter than 5,000 bp (Fig. 1B). LncRNAs were initially classified as intergenic or genic based on their proximity to protein-coding genes (PCGs). Among the genic lncRNAs, 473 (43.7%) were classified as antisense, overlapping a PCG either partially or completely on the opposite strand, and 105 (9.7%) were classified as intronic, being entirely nested within introns. Three putative lncRNAs could not be classified using these criteria.

Among the 1,082 novel lncRNAs, 634 exhibited stage-specific expression patterns (Fig. 2), with the majority being expressed during intermediate-host-related stages: 107 in 1-day sporocysts, 95 in eggs, 88 in miracidia, 85 in 5-day sporocysts, 74 in 32-day sporocysts, 70 in juveniles, 59 in schistosomula, and 56 in cercariae. A similar trend was observed for shared lncRNAs, which were more abundant in intermediate-host stages than in definitive-host stages. These findings are reinforced by the differential transcript expression analysis included in the Supplementary data (Figs 1-2), which shows consistent expression patterns across developmental stages.

*lncRNA-mRNA co-expression* - To investigate putative roles of lncRNAs in stage-related pathways, we performed a co-expression analysis between *S. mansoni* annotated mRNAs and the lncRNAs identified in this project. The transcripts with correlation in expression were clustered in 14 modules, represented by colours. The largest modules were turquoise, followed by blue, and the smallest was light green (Fig. 3A). Each module was correlated with the *S. mansoni* development stages analysed (Fig. 3B). The most related modules per stage were yellow for juveniles, salmon for schistosomula, brown for cercariae, magenta for 32-days sporocysts, and black for eggs.

After defining the co-expression modules, we selected the yellow module for further analysis due to its strong association with the juvenile's stage and its potential to contain biomarkers candidates for schistosomiasis. A network visualisation was constructed for this module, highlighting three hub lncRNAs. These lncRNAs and their co-expressed protein-coding genes (PCGs) were visualised using Cytoscape (Fig. 4A). Among them, the

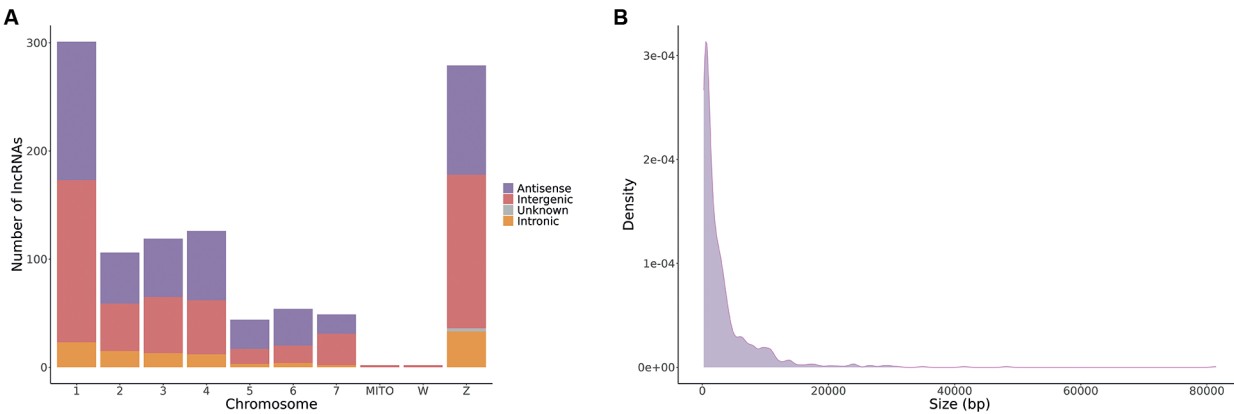

Fig. 1: characterisation of identified lncRNAs. (A) Number of lncRNAs per chromosome, classified as antisense (violet), intergenic (orange), intronic (red), and unknown classification (grey). (B) Length distribution of lncRNAs in base pairs (bp).

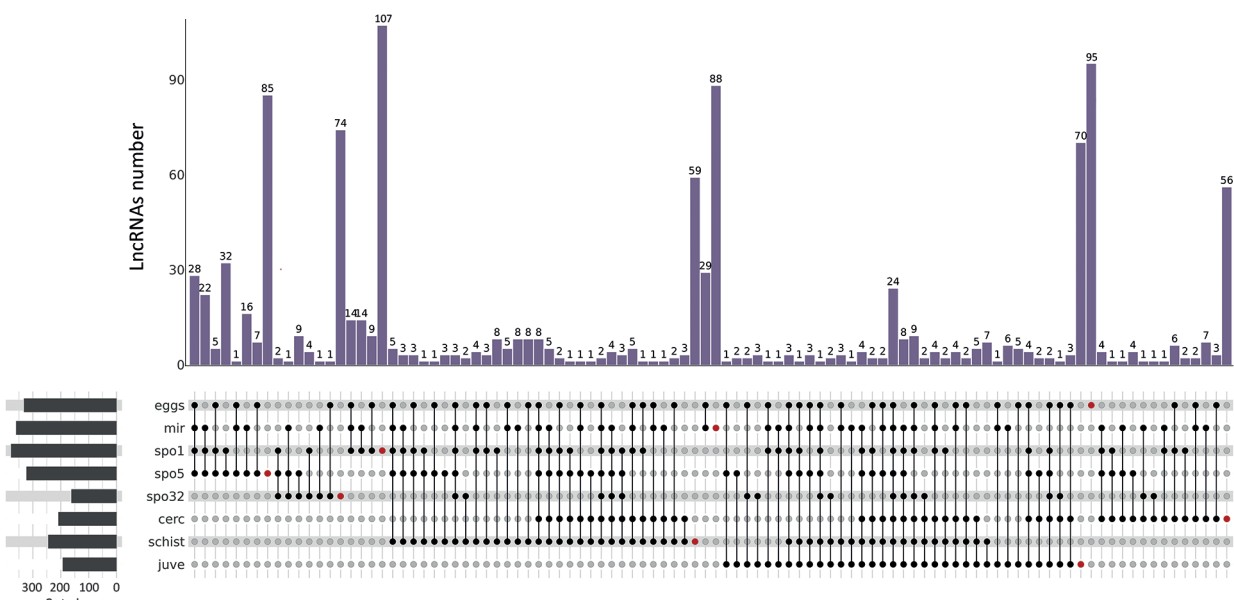

Fig. 2: upSet plot showing the distribution of lncRNAs across *Schistosoma mansoni* developmental stages: eggs, miracidia (mir), sporocyst one (1) day (spo1), five days (spo5), 32 days (spo32), cercariae (cerc), two days schistosomula (schist), and 26 days juveniles (juve). The bars at the top represent the number of lncRNAs (y-axis) shared among the stage combinations indicated by the connected black dots in the lower panel. Red dots indicate unique lncRNAs expressed exclusively in each stage. The set size panel on the left shows the total number of lncRNAs expressed in each developmental stage.

lncRNAs MSTRG.5305.1 transcript showed the highest number of interactions, strongly associated with genes encoding tetraspanins (Smp_154180, Smp_334190, and Smp_346900) and microexon genes (MEGs), including MEG-4.1 (Smp_307220), MEG-32.2 (Smp_123200), and MEG-29 (Smp_243770). The second lncRNA, MSTRG.804.4, was more closely associated with genes involved in peptide metabolism. The third hub, MSTRG.6595.5, demonstrated a more heterogeneous set of connections, including correlations with tetraspanins, MEG-5, membrane-associated proteins, lifeguard protein 4 (implicated in apoptosis), and a non-defined secreted protein. The most significantly enriched GO term for this module was *membrane* (Fig. 4B), consistent with the large number of tetraspanins and other membrane-as-

sociated proteins identified as putative targets of the hub lncRNAs. Additional enriched terms included *lysosome* and *protein metabolic process*, further supporting the involvement of these lncRNAs in membrane dynamics and protein turnover. Additional hub lncRNAs and their respective co-expression sub-networks from other modules are available in the Supplementary data (Figs 3-5).

As these hub lncRNAs were predicted from the juvenile-stage dataset, we experimentally validated their expression in adult worm samples and compared it with cercariae (Fig. 5). Consistent with the bioinformatic data, all three hub lncRNAs were detected in adult worms, with MSTRG.804.4 showing the highest expression level. Little or no expression was observed in cercariae, reproducing the RNA-seq Salmon counts despite

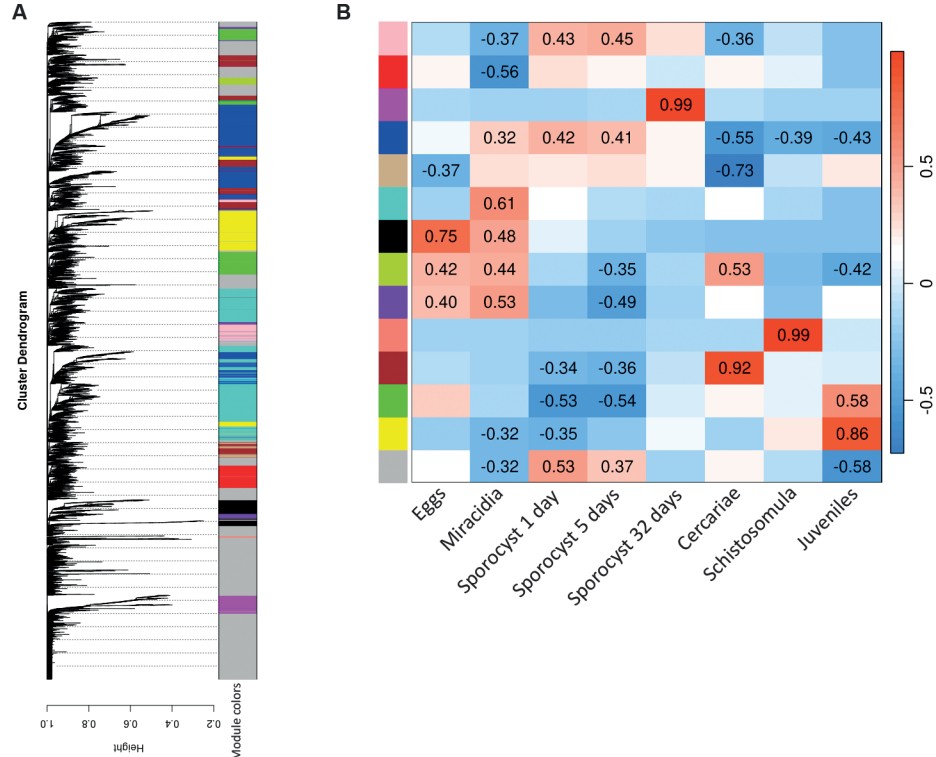

Fig. 3: weighted gene co-expression network analysis (WGCNA) of *Schistosoma mansoni* mRNA and lncRNA. (A) Dendrogram showing 14 lncRNA-mRNA modules clustered according to their expression patterns. Each co-expression module is represented by a distinct colour. (B) Heatmap of the correlation between the modules (rows) and developmental stages of *S. mansoni* (columns). The strength and direction of the association of each module and stage were calculated by a Pearson's correlation, indicated in a scale ranging from -1 (blue) to 1 (red), representing negative and positive correlation, respectively. Only significant values (p < 0.05) were represented.

being obtained from independent samples, in which only MSTRG.6595.5 showed the detection of a few reads in cercariae RNA-seq libraries. The expression differences between adult worms and cercariae were statistically significant for all three lncRNAs (p < 0.05).

## DISCUSSION

LncRNAs are known to play diverse roles in cellular processes and are frequently associated with various physiological and pathological conditions.[24] In parasitic diseases, lncRNAs have been increasingly recognised for their potential involvement in the host-parasite interplay, contributing to parasite development, immune evasion, and adaptation.[9,25,26,27]

Recent advancements in genome assemblies and computational tools have significantly improved the identification of lncRNAs. Despite these technological improvements, our findings show that previously identified lncRNAs remain largely consistent with newly generated datasets, even if not yet fully annotated.[11,12,13,14] This underscores the robustness of earlier discoveries while also emphasising the continued need for refinement of the genome, since it is still incomplete, and lncRNA prediction tools, particularly in non-model organisms.

In this study, most identified lncRNAs were predominantly expressed in intermediate-host developmental stages of *S. mansoni*. This likely reflects a historical bias in transcriptomic studies, which have mainly fo-

cused on stages related to the definitive host.[12,13,14] The limited exploration of intermediate stages is often attributed to technical challenges, such as difficulties in culturing larval forms and separating parasite cells from those of the host.[13,15] However, emerging technologies like single-cell RNA sequencing (scRNA-seq) may help overcome these limitations in the near future. Recent single-cell transcriptomic analyses, such as the study by Morales-Vicente et al.,[28] have already provided valuable insights into lncRNA regulation in *S. mansoni*, revealing tissue-specific expression patterns and identifying potential markers related to the parasite's neural and reproductive systems.

Notably, the larval stages of *S. mansoni* are exposed to more dynamic and stressful environmental conditions compared to adult worms. This may explain the more heterogeneous and stage-specific expression of regulatory molecules such as lncRNAs, suggesting that these transcripts are integral to the parasite's developmental plasticity, as demonstrated in other studies.[12,13,15,29,30]

Although many studies have focused on identifying lncRNAs, functional characterisation remains the major challenge. Computational prediction methods for inferring lncRNA function are still limited, and experimental validation is essential. Co-expression analysis remains the most widely used approach to predict putative lncRNA functions and prioritise candidates for downstream experiments.[9]

**A**

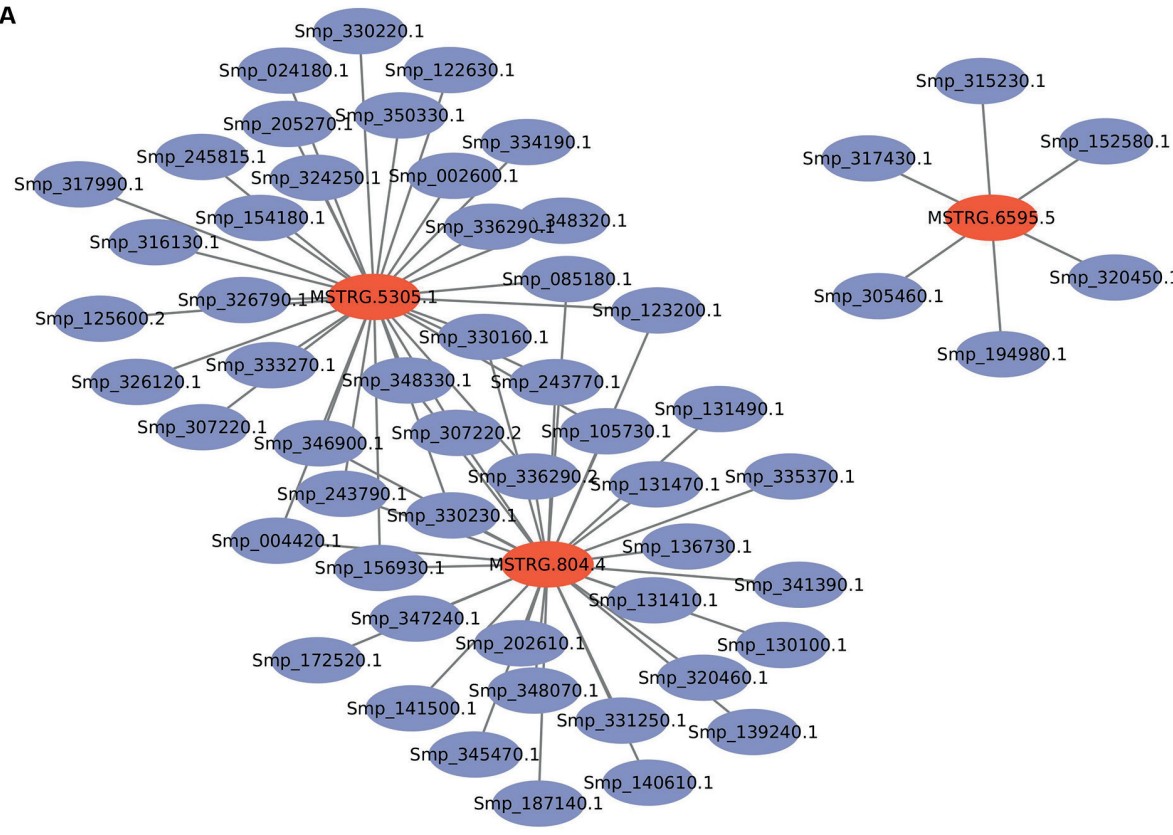

**B**

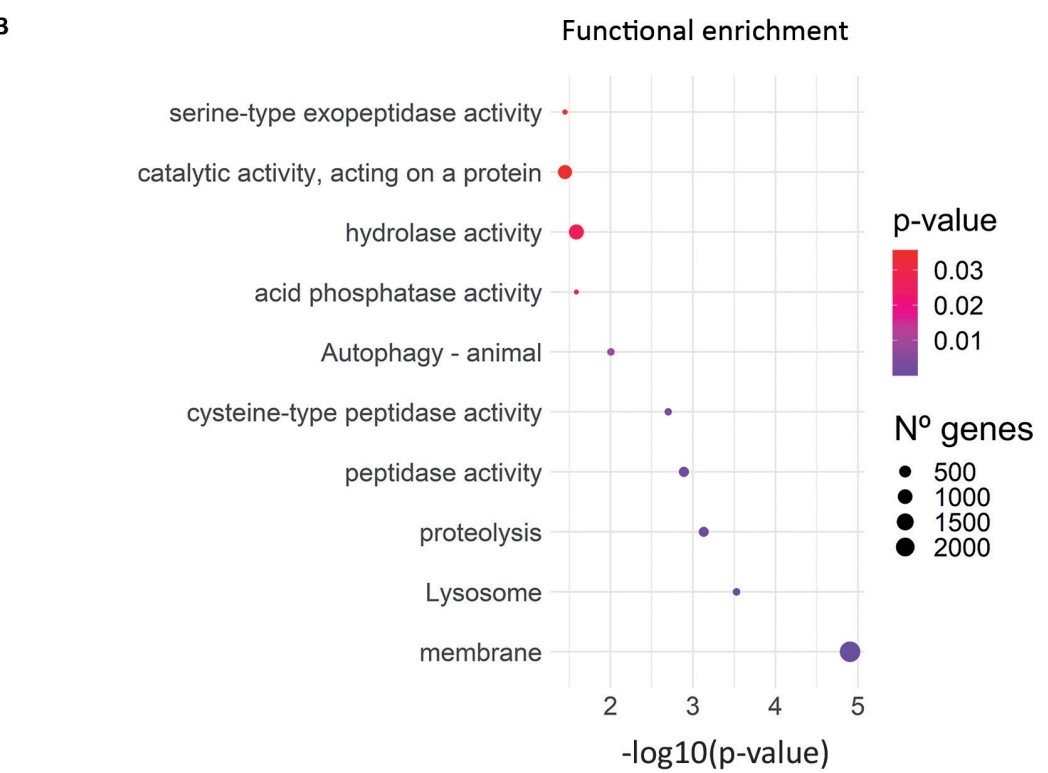

Fig. 4: characterisation of hubs lncRNAs from the yellow module. (A) Subnetwork showing interactions of hub (MM > 0.75 and GS > 0.5) ln-cRNAs (red) and their co-expressed mRNAs (blue) from the yellow module. (B) Gene Ontology (GO) enrichment analysis of the co-expressed mRNAs. The y-axis represents the enriched GO terms, while circle size corresponds to the number of genes associated with each term. The x-axis represents the significance level as -log(p-value), with colours ranging from red (p > 0.03) to violet (p < 0.01).

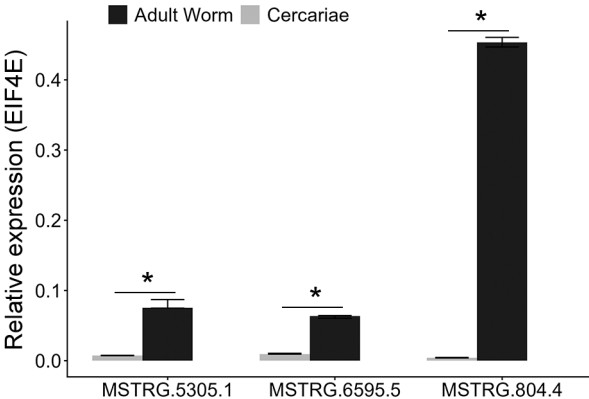

Fig. 5: relative expression of lncRNAs in adult worm and cercariae samples. Expression levels were normalised to the endogenous control gene EIF4E using $2^{-\Delta Cq}$ method, and compared between groups using the Mann-Whitney test (*$p < 0.05$).

In our co-expression network analysis, we focused on the yellow module, which is strongly associated with juvenile stage. Among its hub lncRNAs, MSTRG.5305.1 emerged as a particularly promising candidate due to its strong co-expression with genes encoding tetraspanins and members of the micro-exon gene (MEG) family. Importantly, the experimental validation of selected hub lncRNAs confirmed their stage-specific expression patterns, supporting the biological reproducibility of our computational predictions and reinforcing their potential as targets for future functional assays.

Tetraspanins are integral components of the schistosome tegument and are involved in maintaining structural integrity and facilitating immune evasion.[31,32,33] These proteins are recognised by IgG1 and IgG3 antibodies only in resistant individuals, and vaccination with recombinant Sm-TSP-2 has been shown to reduce worm and egg burdens in mice.[33] RNAi-mediated silencing of Sm-tsp-1 and Sm-tsp-2 tetraspanin mRNAs leads to tegumental disruption and decreased of parasite viability for murine models, although the tegumental turnover was observed to increase in *in vitro* assays.[34] More recently, a phase I clinical trial of the Sm-TSP-2 vaccine in humans demonstrated that it is safe and effective at inducing IgG responses.[35]

MEGs are a group of genes characterised by short exons (3-81 bp) interspersed with long introns. These genes are capable of generating numerous protein isoforms through alternative splicing, a feature believed to contribute to immune evasion.[4] Many MEGs also possess a signal peptide at the 5′ end, suggesting they are secreted. Although MEG-32.2 and MEG-29 remain poorly characterised, MEG-4.1 is a known esophageal-secreted protein involved in blood processing.[36,37] Several MEGs have been evaluated as vaccine candidates, though they generally exhibit low efficacy.[36,38,39] The observed correlation between MSTRG.5305.1 and these MEGs and tetraspanins highlights this lncRNA as a compelling target for further functional investigation.

Ultimately, identifying lncRNAs that regulate well-characterised parasite genes offers valuable insights into novel molecular pathways. These findings demonstrate the promising involvement of lncRNAs in essential processes for juvenile survival. This valuable insight will need to be addressed in future functional validation studies, which may pave the way for the development of new diagnostics, therapeutic strategies, and transmission-blocking interventions against schistosomiasis.

## AUTHORS' CONTRIBUTION

FAR participates of all data analysis and writing; CC collaborates with bioinformatic analysis; CG and RG have conceived the project, contributed with data interpretation, writing and reviewing the manuscript. The authors declare that there are no conflicts of interest regarding the publication of this paper.

## DATA AVAILABILITY

The contents underlying the research text are included in the manuscript.

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

# OPEN PEER REVIEW

Memórias do IOC thanks the anonymous reviewers for their contribution to the peer review of this work.

**FIRST REVIEW ROUND**

**REVIEWERS' COMMENTS**

**REVIEWER #1**

Dear authors,

I have completed the review of your manuscript entitled " Comprehensive Identification and Co-expression Analysis of Long Non-coding RNAs Across Eight Timepoints of Schistosoma mansoni Life Cycle" submitted Memórias do Instituto Oswaldo Cruz. I commend you for your work and would like to provide some feedback.

Strengths:

Overall, this manuscript is well-written, comprehensive and provides a thorough analysis of the long non-coding RNAs across different S. mansoni life stages using publicly available RNA-seq data.

Suggestions for improvement:

- WGCNA analysis: The analysis is restricted to a single developmental stage (adult worms), while the manuscript emphasizes the need for broader investigations across the parasite's life cycle. Extending the WGCNA to additional stages would strengthen the study.

- Differential expression analysis: Including a differential expression analysis across life stages, accompanied by a heatmap, would greatly enhance the clarity of the results and highlight biologically meaningful patterns.

- Functional enrichment: Have the authors attempted to perform enrichment analysis of co-expressed or differentially expressed genes/lncRNAs? Such an analysis could provide important functional insights and strengthen the biological interpretation.

- Figures: Highlighting stage-specific groups in Figure 2 (e.g., using a red dot or distinct color markers) would help readers quickly identify key findings.

- Figure Captions: Please revise figure captions to make them more informative and self-explanatory, providing sufficient context without requiring readers to refer extensively to the main text.

Minor comments:

- Page 2, Line 48 (Objectives): I suggest rewriting "identify and characterize novel lncRNAs", as the current phrasing does not fully reflect the objectives of the paper.

- Page 3, Lines 20 – 28 (Conclusions): Please rewrite the conclusion based on the study's findings. At present, it reads more like a general conclusion derived from literature.

- Page 3, Line 48: Please correct for Schistosoma mansoni, Schistosoma haematobium, and Schistosoma japonicum, as it is the first time these species are mentioned.

- Page 4, Lines 41 - 53:  Please add an appropriated reference.

- Page 6, Line 7: Please, replace "8" with "eight".

- Page 6, Line 11 and 20 – 25: Please rewrite these sentences to clarify that the datasets were generated by other authors from original material. For example:

- "All RNA-seq datasets originate from RNA extracted from mixed samples…"

- "The eggs, 32-day sporocyst, and juvenile data originate from samples originally collected and purified from host tissues, whereas the other stages were cultivated and transformed in vitro.".

- Page 7, Line 27: Please specify which BLAST search was used.

- Page 8, Line 46: I suggest replacing the word "generating".

- Figure 1: Is there any lncRNA with ~80,000 bp?

- Figure 2: As noted above, highlighting stage-specific groups would improve readability.

- Page 12, Line 16: Replace "unfinished" with a more precise term (e.g., "incomplete").

- Page 12, Line 17: Replace "in" with "for".

- Page 12, Line 34: The authors could strengthen the discussion by integrating findings from Morales et al. (doi: 10.3389/fgene.2022.924877), which provide additional insights from single-cell data.

- Reference 5: This is not the correct reference for the most recent genome update; please revise accordingly.

**AUTHORS' RESPONSE TO THE REVIEWERS**

Dear Dr. Adeilton Brandão
Handling Editor, Memórias do Instituto Oswaldo Cruz

Regarding the manuscript MIOC-2025-0114 entitled "Comprehensive Identification and Co-expression Analysis of Long Non-coding RNAs Across Eight Timepoints of Schistosoma mansoni Life Cycle", we have

carefully reviewed and revised the manuscript according to the reviewers' suggestions. Below, we address each point raised, indicating where changes were made in the text. We would like to take this opportunity to sincerely thank the reviewer for their thorough and insightful feedback, which greatly facilitated our revision process and undoubtedly enhanced the quality of our manuscript.

ANSWER TO REVIEWER

1. WGCNA analysis: The analysis is restricted to a single developmental stage (adult worms), while the manuscript emphasizes the need for broader investigations across the parasite's life cycle. Extending the WGCNA to additional stages would strengthen the study.

Answer: We thank the reviewer for this valuable comment. WGCNA was performed for all developmental stages and hub lncRNAs were identified within subnetworks for some of them. The juvenile stage was retained for detailed analysis due to its robust co-expression patterns and potential relevance for subsequent functional screening. For transparency, we have now included supplementary figures showing the co-expression results for other stages in which hub lncRNAs were detected (see Supplementary Data).

2. Differential expression analysis: Including a differential expression analysis across life stages, accompanied by a heatmap, would greatly enhance the clarity of the results and highlight biologically meaningful patterns.

We thank the reviewer for this valuable comment. We have included a supplementary figure presenting a heatmap across the different developmental stages. Importantly, Figure 2 emphasizes the features we consider most relevant and informative for the reader, particularly alongside the functional enrichment analysis added in the revised manuscript. A central aim of our group is to prioritize molecules with the highest potential for future experimental validation. For long non-coding RNAs, such validation could involve approaches like CRISPR-mediated knockdown or activation, antisense oligonucleotides, or other emerging strategies to modulate lncRNA expression and assess their biological function.

3. Functional enrichment: Have the authors attempted to perform enrichment analysis of co-expressed or differentially expressed genes/lncRNAs? Such an analysis could provide important functional insights and strengthen the biological interpretation.

Answer: We thank the reviewer for this important comment. Functional enrichment analyses (GO, KEGG and REAC) were performed for each module. The results are now included in the Results section (pages 12-13) and Figure 4, providing additional functional insights and supporting the biological interpretation of our findings.

4. Figures: Highlighting stage-specific groups in Figure 2 (e.g., using a red dot or distinct color markers) would help readers quickly identify key findings.

Answer: Thank you for the valuable suggestion. Figure 2 has been revised to highlight stage-specific groups using distinct colors, improving clarity and readability. Please see page 11 and Figure 2.

5. Figure Captions: Please revise figure captions to make them more informative and self-explanatory, providing sufficient context without requiring readers to refer extensively to the main text.

Answer: We thank the reviewer for this valuable suggestion. The figure captions have been thoroughly revised to make them more informative and self-explanatory, providing sufficient context so that the reader can understand the figures without extensively referring to the main text.

Minor comments:

1. Page 2, Line 48 (Objectives): I suggest rewriting "identify and characterize novel lncRNAs", as the current phrasing does not fully reflect the objectives of the paper.

Answer: Thank you for your comment. We have reviewed the terms indicated and modified them for a more concise reading. Please see page 2 line 48.

2. Page 3, Lines 20 – 28 (Conclusions): Please rewrite the conclusion based on the study's findings. At present, it reads more like a general conclusion derived from literature.

Answer: Thank you for your comment. We have reviewed the terms indicated and modified them for a more concise reading. Please see page 3 line 20-28.

3. Page 3, Line 48: Please correct for Schistosoma mansoni, Schistosoma haematobium, and Schistosoma japonicum, as it is the first time these species are mentioned.

Answer: Thank you for your comment. We have reviewed the terms indicated and modified them for a more concise reading. Please see page 3 line 55-58.

4.- Page 4, Lines 41 - 53: Please add an appropriated reference.

Answer: Thank you for your comment. We have reviewed the terms indicated and modified them for a more concise reading. Please see page 5 line 20.

5. Page 6, Line 7: Please, replace "8" with "eight".

Answer: Thank you for your comment. We have reviewed the terms indicated and modified them for a more concise reading. Please see page 6 line 18.

6. Page 6, Line 11 and 20 – 25: Please rewrite these sentences to clarify that the datasets were generated by other authors from original material. For example:

- "All RNA-seq datasets originate from RNA extracted from mixed samples…"
- "The eggs, 32-day sporocyst, and juvenile data originate from samples originally collected and purified from host tissues, whereas the other stages were cultivated and transformed in vitro."

Answe: Thank you for your comment. We have reviewed the terms indicated and modified them for a more concise reading. Please see page 6 line 23-42.

7.Page 7, Line 27: Please specify which BLAST search was used.

Answer: Thank you for your comment. We have reviewed the terms indicated and modified them for a more concise reading. Please see page 7 line 46.

8. Page 8, Line 46: I suggest replacing the word "generating".

Answer: Thank you for your comment. We have reviewed the terms indicated and modified them for a more concise reading. Please see page 10 line 27.

9. Figure 1: Is there any lncRNA with ~80,000 bp?

Answer: Yes. Despite their rare occurrence, some authors consider the length range up to 90 kbp or above (https://doi.org/10.1016/j.lfs.2021.119152), as KCNQ1OT1 (https://doi.org/10.1016/j.biopha.2023.115876).

10. Figure 2: As noted above, highlighting stage-specific groups would improve readability.

Answer: Thank you for your comment. We have reviewed the terms indicated and modified them for a more concise reading.

11. Page 12, Line 16: Replace "unfinished" with a more precise term (e.g., "incomplete").

Thank you for your comment. We have reviewed the terms indicated and modified them for a more concise reading. Please see page 15 line 29.

12. Page 12, Line 17: Replace "in" with "for".

Answer: Thank you for your comment. We have reviewed the terms indicated and modified them for a more concise reading.

13. Page 12, Line 34: The authors could strengthen the discussion by integrating findings from Morales et al. (doi: 10.3389/fgene.2022.924877), which provide additional insights from single-cell data.

Answer: Thank you for your comment. We have reviewed the terms indicated and modified them for a more concise reading. Please see page 15 line 50-59.

14. - Reference 5: This is not the correct reference for the most recent genome update; please revise accordingly.

Answer: Thank you for your comment. We have reviewed the terms indicated and modified them for a more concise reading.

Additionally, we included RT-qPCR data for hub lncRNAs from the yellow module in adult worm and cercariae RNA samples. This analysis was performed while the manuscript was under review and adds substantial robustness to our findings.

Cordially,
Prof. Dr. Renata Guerra-Sá
(Corresponding Author) and Dra. Arêdes-Rocha

## SECOND REVIEW ROUND

### REVIEWERS' COMMENTS

**REVIEWER #1**

Dear authors,
I have completed the review of your manuscript entitled "Comprehensive Identification and Co-expression Analysis of Long Non-coding RNAs Across Eight Timepoints of Schistosoma mansoni Life Cycle" submitted to Memórias do Instituto Oswaldo Cruz. I would like to commend the authors for your thorough work, the careful revision following my previous comments, and the inclusion of RT-qPCR data, which indeed adds substantial robustness to your findings.

Suggestions for improvement:
- RT-qPCR: Please include the primers sequences for the endogenous control (Eif4e) and provide a reference for the analysis method used ($2^{-\Delta Cq}$ method). I also suggest referring as an endogenous control rather than a reference gene , as this terminology is more consistent with current RT-qPCR standards.
- Statistical Analysis: Please add the R packages used for the statistical analyses.
Minor comments:
- Page 6, Line 58: Please, correct "Schistsoma japonicum" to "Schistosoma japonicum".
- Page 12, Lines 13-16: I suggest rephrasing "RNA quantity and purity were assessed using a spectrophotometer and agarose gel electrophoresis" for "RNA quantity and purity were assessed using a spectrophotometer, and RNA integrity was verified by agarose gel electrophoresis".
- Figure 5: The asterisks mentioned in the figure caption appear to be missing from the figure itself. Please check and correct this.

