## [Reviewer Report · FIRST REVIEW ROUND - REVIEWERS COMMENTS]

## REVIEWER #1

Dear authors,

I have completed the review of your manuscript entitled “Comprehensive Identification and Co-expression Analysis of Long Non-coding RNAs Across Eight Timepoints of *Schistosoma mansoni* Life Cycle” submitted Memórias do Instituto Oswaldo Cruz.

I commend you for your work and would like to provide some feedback.

**Strengths:**

Overall, this manuscript is well-written, comprehensive and provides a thorough analysis of the long non-coding RNAs across different *S. mansoni* life stages using publicly available RNA-seq data.

**Suggestions for improvement:**

- WGCNA analysis: The analysis is restricted to a single developmental stage (adult worms), while the manuscript emphasizes the need for broader investigations across the parasite’s life cycle. Extending the WGCNA to additional stages would strengthen the study.

- Differential expression analysis: Including a differential expression analysis across life stages, accompanied by a heatmap, would greatly enhance the clarity of the results and highlight biologically meaningful patterns.

- Functional enrichment: Have the authors attempted to perform enrichment analysis of co-expressed or differentially expressed genes/lncRNAs? Such an analysis could provide important functional insights and strengthen the biological interpretation.

- Figures: Highlighting stage-specific groups in Figure 2 (e.g., using a red dot or distinct color markers) would help readers quickly identify key findings.

- Figure Captions: Please revise figure captions to make them more informative and self-explanatory, providing sufficient context without requiring readers to refer extensively to the main text.

**Minor comments:**

- Page 2, Line 48 (Objectives): I suggest rewriting “identify and characterize novel lncRNAs”, as the current phrasing does not fully reflect the objectives of the paper.

- Page 3, Lines 20 – 28 (Conclusions): Please rewrite the conclusion based on the study’s findings. At present, it reads more like a general conclusion derived from literature.

- Page 3, Line 48: Please correct for *Schistosoma mansoni*, *Schistosoma haematobium*, and *Schistosoma japonicum*, as it is the first time these species are mentioned.

- Page 4, Lines 41 - 53: Please add an appropriated reference.

- Page 6, Line 7: Please, replace “8” with “eight”.

- Page 6, Line 11 and 20 – 25: Please rewrite these sentences to clarify that the datasets were generated by other authors from original material.

- Page 7, Line 27: Please specify which BLAST search was used.

- Page 8, Line 46: I suggest replacing the word “generating”.

- Figure 1: Is there any lncRNA with ~80,000 bp?

- Figure 2: As noted above, highlighting stage-specific groups would improve readability.

- Page 12, Line 16: Replace “unfinished” with a more precise term (e.g., “incomplete”).

- Page 12, Line 17: Replace “in” with “for”.

- Page 12, Line 34: The authors could strengthen the discussion by integrating findings from Morales et al. (doi: 10.3389/fgene.2022.924877), which provide additional insights from single-cell data.

- Reference 5: This is not the correct reference for the most recent genome update; please revise accordingly.

## AUTHORS' RESPONSE TO THE REVIEWERS

Dear Dr. Adeilton Brandão

Handling Editor, Memórias do Instituto Oswaldo Cruz

Regarding the manuscript **MIOC-2025-0114** entitled “Comprehensive Identification and Co-expression Analysis of Long Non-coding RNAs Across Eight Timepoints of *Schistosoma mansoni* Life Cycle”, we have carefully reviewed and revised the manuscript according to the reviewers’ suggestions.

Below, we address each point raised, indicating where changes were made in the text.

We would like to take this opportunity to sincerely thank the reviewer for their thorough and insightful feedback, which greatly facilitated our revision process and undoubtedly enhanced the quality of our manuscript.

**ANSWER TO REVIEWER**

**1. WGCNA analysis:** The analysis is restricted to a single developmental stage (adult worms), while the manuscript emphasizes the need for broader investigations across the parasite’s life cycle. Extending the WGCNA to additional stages would strengthen the study.

**Answer:** We thank the reviewer for this valuable comment. WGCNA was performed for all developmental stages and hub lncRNAs were identified within subnetworks for some of them. The juvenile stage was retained for detailed analysis due to its robust co-expression patterns and potential relevance for subsequent functional screening. For transparency, we have now included supplementary figures showing the co-expression results for other stages in which hub lncRNAs were detected (see Supplementary data).

**2. Differential expression analysis:** Including a differential expression analysis across life stages, accompanied by a heatmap, would greatly enhance the clarity of the results and highlight biologically meaningful patterns.

**Answer:** We thank the reviewer for this valuable comment. We have included a supplementary figure presenting a heatmap across the different developmental stages. Importantly, Figure 2 emphasizes the features we consider most relevant and informative for the reader, particularly alongside the functional enrichment analysis added in the revised manuscript.

**3. Functional enrichment:** Have the authors attempted to perform enrichment analysis of co-expressed or differentially expressed genes/lncRNAs?

**Answer:** We thank the reviewer for this important comment. Functional enrichment analyses (GO, KEGG and REAC) were performed for each module. The results are now included in the Results section (pages 12-13) and Figure 4, providing additional functional insights and supporting the biological interpretation of our findings.

**4. Figures:** Highlighting stage-specific groups in Figure 2 (e.g., using a red dot or distinct color markers) would help readers quickly identify key findings.

**Answer:** Thank you for the valuable suggestion. Figure 2 has been revised to highlight stage-specific groups using distinct colors, improving clarity and readability. Please see page 11 and Figure 2.

**5. Figure Captions:** Please revise figure captions to make them more informative and self-explanatory, providing sufficient context without requiring readers to refer extensively to the main text.

**Answer:** We thank the reviewer for this valuable suggestion. The figure captions have been thoroughly revised to make them more informative and self-explanatory.

...

Cordially,

Prof. Dr. Renata Guerra-Sá (Corresponding Author) and Dra. Arêdes-Rocha

---

## [Reviewer Report · SECOND REVIEW ROUND - REVIEWERS COMMENTS]

## REVIEWER #1

Dear authors,

I have completed the review of your manuscript entitled “Comprehensive Identification and Co-expression Analysis of Long Non-coding RNAs Across Eight Timepoints of *Schistosoma mansoni* Life Cycle” submitted to Memórias do Instituto Oswaldo Cruz.

I would like to commend the authors for your thorough work, the careful revision following my previous comments, and the inclusion of RT-qPCR data, which indeed adds substantial robustness to your findings.

**Suggestions for improvement:**

- RT-qPCR: Please include the primers sequences for the endogenous control (Eif4e) and provide a reference for the analysis method used (2^−ΔCq method). I also suggest referring as an endogenous control rather than a reference gene , as this terminology is more consistent with current RT-qPCR standards.

- Statistical Analysis: Please add the R packages used for the statistical analyses.

**Minor comments:**

- Page 6, Line 58: Please, correct “*Schistsoma japonicum*” to “*Schistosoma japonicum*”.

- Page 12, Lines 13-16: I suggest rephrasing “RNA quantity and purity were assessed using a spectrophotometer and agarose gel electrophoresis” for “RNA quantity and purity were assessed using a spectrophotometer, and RNA integrity was verified by agarose gel electrophoresis”.

- Figure 5: The asterisks mentioned in the figure caption appear to be missing from the figure itself. Please check and correct this.